# Novel 2-alkythio-4-chloro-*N*-[imino(heteroaryl)methyl]benzenesulfonamide Derivatives: Synthesis, Molecular Structure, Anticancer Activity and Metabolic Stability

**DOI:** 10.3390/ijms24119768

**Published:** 2023-06-05

**Authors:** Beata Żołnowska, Jarosław Sławiński, Mariusz Belka, Tomasz Bączek, Jarosław Chojnacki, Anna Kawiak

**Affiliations:** 1Department of Organic Chemistry, Medical University of Gdańsk, Al. Gen. J. Hallera 107, 80-416 Gdańsk, Poland; jaroslaw.slawinski@gumed.edu.pl; 2Department of Pharmaceutical Chemistry, Medical University of Gdańsk, Al. Gen. J. Hallera 107, 80-416 Gdańsk, Poland; mariusz.belka@gumed.edu.pl (M.B.); tbaczek@gumed.edu.pl (T.B.); 3Department of Inorganic Chemistry, Gdańsk University of Technology, ul. Narutowicza 11/12, 80-233 Gdańsk, Poland; jaroslaw.chojnacki@pg.edu.pl; 4Department of Biotechnology, Intercollegiate Faculty of Biotechnology, University of Gdańsk and Medical University of Gdańsk, ul. Abrahama 58, 80-307 Gdańsk, Poland

**Keywords:** benzenesulfonamide, imidazole, synthesis, anticancer, apoptosis, metabolic stability, cell cycle flow cytometry analysis

## Abstract

A series of novel 2-alkythio-4-chloro-*N*-[imino-(heteroaryl)methyl]benzenesulfonamide derivatives, **8**–**24**, were synthesized in the reaction of the *N*-(benzenesulfonyl)cyanamide potassium salts **1**–**7** with the appropriate mercaptoheterocycles. All the synthesized compounds were evaluated for their anticancer activity in HeLa, HCT-116 and MCF-7 cell lines. The most promising compounds, **11**–**13**, molecular hybrids containing benzenesulfonamide and imidazole moieties, selectively showed a high cytotoxic effect in HeLa cancer cells (IC_50_: 6–7 μM) and exhibited about three times less cytotoxicity against the non-tumor cell line HaCaT cells (IC_50_: 18–20 μM). It was found that the anti-proliferative effects of **11**, **12** and **13** were associated with their ability to induce apoptosis in HeLa cells. The compounds increased the early apoptotic population of cells, elevated the percentage of cells in the sub-G1 phase of the cell cycle and induced apoptosis through caspase activation in HeLa cells. For the most active compounds, susceptibility to undergo first-phase oxidation reactions in human liver microsomes was assessed. The results of the in vitro metabolic stability experiments indicated values of the factor t_½_ for **11**–**13** in the range of 9.1–20.3 min and suggested the hypothetical oxidation of these compounds to sulfenic and subsequently sulfinic acids as metabolites.

## 1. Introduction

Cancer is a major public health problem and a leading cause of death worldwide which caused nearly 10 million deaths in 2020 [1]. Basic and clinical research are still needed to increase our knowledge about cancer and accelerate progress in the fight against it. Breast, cervical and colorectal cancers are the most common female cancer types worldwide. In women, the incidence rates of breast cancer far exceed those of other cancers in both transitioned (55.9 per 100,000) and transitioning (29.7 per 100,000) countries, followed by those of colorectal cancer (20 per 100,000) in transitioned countries and of cervical cancer (18.8 per 100,000) in transitioning countries [2].

One of the most important strategies in the search for chemotherapeutics is the approach based on combining in one molecule building blocks, fragments of known drugs, leading structures or “hit” structures [3,4,5,6]. The conjugation of two pharmacophores into a single chemical entity called a hybrid aims at achieving a synergistic effect with increased efficacy compared to the starting compounds [7,8]. The pharmacophores can be combined using three basic types of conjugation leading to linked, fused and merged hybrids [9,10,11,12]. Among the advantages of the molecular hybrid are higher activity due to its effect on many molecular targets, minimization of drug resistance and side effects, and improvement of the pharmacokinetic properties [13]. From the molecular design point of view, the combination of two pharmacophores into a single molecule represents one of the methods that can be adopted for the synthesis of new anticancer molecules [14].

The United States Food and Drug Administration (FDA) databases reveals the structural importance of nitrogen-based heterocycles in designing pharmaceuticals. Closely 75% of unique small-molecule drugs contain a nitrogen heterocycle [15]. Imidazoles belong to the most frequently appearing nitrogen heterocycles in small-molecule drugs. These ring systems containing two nitrogen atoms influence a wide range of biological activities such as anticancer [16], antibacterial [17], antiviral [18], antiepileptic [19], antitubercular [20] and antifungal activities [21]. Among anticancer imidazoles, nilotinib is known as a second-generation tyrosine kinase inhibitor, widely used in the treatment of Chronic Myeloid Leukemia (CML) [22], and dacarbazine as an alkylating agent exhibiting antitumor activity by DNA methylation in colorectal cancers [23].

1,2,4-Triazoles are attractive targets for research due to a number of biological activities including antimicrobial [24], antifungal [25], anti-inflammatory [26], antitubercular [27], antiviral [28], analgesic [29], and anticancer activities [30]. Literature reports present important chemotherapeutic agents, such as anastrozole, letrozole and vorozole, containing the 1,2,4-triazole ring, which are currently used in the treatment of breast cancer, with a mechanism of action related to aromatase inhibition [31].

Various sulfonamide compounds are used in anticancer therapies (Figure 1). Sulfonamide drugs such as fedratinib (Janus kinase inhibitor) [32], pazopanib (VEGFR growth receptor inhibitor) [33], dabrafenib [34], and vemurafenib (NCT04302025) (which are inhibitors of BRAF V600 protein) lead to the inhibition of cancer cell division. For example, vemurafenib is used as a targeted monotherapy to treat melanoma with a mutation in the BRAF V600 protein [35]. Venetoclax [36], navitoclax [37] and tasisulam [38] are drugs that induce apoptosis.

Apoptosis is a genetically regulated and evolutionarily conserved process with key roles in cellular proliferation and tissue homeostasis [39]. Most anticancer treatments such as chemotherapy, radiotherapy and immunotherapy primarily aim to activate apoptosis; thus, the ability to induce apoptosis in cancer cells is a highly desirable feature of chemotherapeutics agents [40]. The apoptosis-inducing activity of molecules can be investigated through cytometric analysis of DNA fragmentation, loss of mitochondrial membrane potential (Δψm), phosphatidylserine translocation to the outer leaflet of the cell membrane and caspase activation, as well as by the microscopic observation of morphological changes in tumor cells after treatment with anticancer agents.

In our research, we designed molecular fused hybrids combining a benzenesulfonamide fragment, a heterocyclic system such as imidazole, 1,2,4-triazole and benzimidazole and an imine group as a linker (Figure 1, Formula A). The structures of the compounds were also diversified with substituents at position 2 of benzenesulfonamide to investigate their influence on the hybrids’ cytotoxic activity. Screening tests were performed in breast, cervical and colorectal cancer cell lines which represent the most problematic cancer diseases in women according to statistics. A path leading to cancer cell death was investigated by staining cells with Annexin V-PE and 7-AAD and monitoring caspase activation and cell cycle distribution, applying flow cytometry. It is known that achieving the chemical and physical stability of drugs is essential to ensure their quality and safety. We carried out metabolic stability studies to predict the number and types of metabolites formed in a typical test system, i.e., liver microsomes, as well as the potential of the designed compounds as drug candidates.

## 2. Results and Discussion

### 2.1. Chemistry

The starting potassium salts **1**–**7** were obtained according to the reported procedures for the preparation of N-(benzenesulfonyl)cyanamide potassium salts [41,42,43,44]. The novel 2-alkythio-4-chloro-N-[imino-(heteroaryl)methyl]benzenesulfonamide derivatives **8**–**24** were synthesized by the reaction of the N-(benzenesulfonyl)cyanamide potassium salts **1**–**7** with 1-methyl-1H-imidazole-2-thiol or 4-methyl-4H-1,2,4-triazole-3-thiol or 1H-benzo[d]imidazole-2-thiol using dry toluene or p-dioxane as a solvent (Figure 1).

The structures of the final compounds **8**–**24** were confirmed by IR, ^1^H NMR and ^13^C NMR spectroscopy. The IR spectra showed the typical stretching vibration of the NH group at nearly 3300 cm^−1^ and the presence of two bands at approximately 1630 and 1540 cm^−1^, corresponding to C=C and C=N stretching. Moreover, the sulfonyl group was identified by bands from S=O stretching (asymmetric and symmetric) at approximately 1370 and 1140 cm^−1^. The appearance of NH signals at 10.22–11.29 and 8.73–8.83 ppm in the ^1^H NMR spectra proved the presence of the SO_2_NH and C=NH groups, while singlets at approximately 2.32–2.37, 4.01–4.82, 7.44–7.74 and 7.4–8.06 ppm confirmed a 2-alkylthio-5-methylbenzenesulfonamide scaffold, i.e., the presence of CH_3_, CH_2_, H-3 and H-6. Moreover, heterocyclic rings attached to a sulfur atom showed specific signals such as doublets at 7.28–7.40 and 7.38–7.59 ppm, describing H-4 and H-5 protons in imidazole, or singlets at 8.73–8.75 ppm, belonging to the H-5 proton in 1,2,4-triazole. Singlets derived from benzimidazole NH protons appeared at 13.58–13.65 ppm.

Compound **10** crystallized in the space group P2_1_/c with one molecule in the asymmetric unit. Details on data collection, structure solution and refinement are reported in Appendix A. A molecular view is presented in Figure 2. In the solid state, the molecule is deprotonated in the sulfonamidic part, and the hydrogen atoms H2a and H2b are located on the amine nitrogen atom N2. They form hydrogen bonds with the neighbor O1 and S3 (for details, see Appendix A). No distinct electron density peak was found in the vicinity of the sulfonamide nitrogen atom N1. Additionally, the CF_3_ group was found disordered over two positions, with a site occupation factor of 0.58(2)/0.42(2).

### 2.2. Screening for Anticancer Agents

Compounds **8**–**24** were evaluated in vitro for their cytotoxic effect against three human cancer cell lines, i.e., HeLa (cervical cancer), HCT-116 (colon cancer) and MCF-7 (breast cancer), and the non-tumor cell line HaCaT using the MTT assay after 72 h of incubation. The results of tests are presented in Table 1 as IC_50_, indicating the concentration required for 50% inhibition of cell viability. Compounds **8**–**10** and **14**–**18** are not presented in the Table 1 because of their low potency (IC_50_ > 100 µM). 

As shown in Table 1, the HeLa cell line exhibited the highest susceptibility toward compounds **11**–**13** (IC_50_; 6–7 µM) representing a series of 3-methyl-2-thioxo-2,3-dihydro-1*H*-imidazole derivatives. It is worth noting that the HCT-116 line was noticeably susceptible to the 2-thioxo-2,3-dihydro-1H-benzo[*d*]imidazole derivatives **19**–**24** (IC_50_; 17–36 µM), but the most active compound was the imidazole derivative **11**, with an IC_50_ value of 11 µM. The MCF-7 cell line provided the weakest response to the cytotoxic effect of the tested compounds, with only the 2-thioxo-2,3-dihydro-1*H*-benzo[*d*]imidazole derivatives **19**–**24** exhibiting an IC_50_ in the range of 49–82 µM. Unfortunately, the 5-thioxo-4,5-dihydro-1*H*-1,2,4-triazole derivatives **15**–**18** had no effect on cancer cell viability. 

We also performed an assay on the non-tumor cell line HaCaT (immortalized human keratinocytes) to assess if the effect of **11**–**13** was selective toward HeLa cells or resulted from a more general toxic activity. The test indicated that the compounds showed selectivity toward cancer cells. Values of IC_50_ in the range of 18–20 µM for HaCaT cells indicated about three times less toxicity than that for HeLa cells. What is important, the cytotoxicity against non-cancerous cells of **11**–**13** was significantly lower than that of the reference drug cisplatin, which is a common drug used for cervical cancer treatment as a cell cycle non-specific drug in the clinic.

### 2.3. Apoptosis

#### 2.3.1. Cytotoxic Activity

The cytotoxic activity of **11**–**13** was determined in a time-dependent manner with the MTT assay (Figure 3).

HeLa cells were treated with **11**, **12** and **13** in the concentration range of 0–20 µM. After 24 h of treatment, the IC_50_ values were not reached by the compounds **11** and **12**, whereas for compound **13**, the IC_50_ value was reached at the concentration of 18 μM. After 48 h, the IC_50_ values for compounds **11**, **12** and **13** were 6, 7 and 6 µM, respectively. Further treatment with the compounds did not increase their cytotoxic activity.

#### 2.3.2. Apoptosis Induction

In order to determine whether the anti-proliferative effects of **11**, **12** and **13** were associated with their ability to induce apoptosis in HeLa cells, the induction of phosphatidylserine externalization by compounds **11**, **12** and **13** was examined by flow cytometric analysis. The cells were treated with 2.5, 5 and 10 µM concentrations of **11**, **12** and **13** for 24 and 48 h and stained with Annexin V-PE and 7-AAD. The results shown in Figure 4 indicated that compounds **11**–**13** induced apoptosis in a concentration- and time-dependent manner. After 24 h of treatment, a significant increase in the early apoptotic population of cells was visible starting from the concentration of 5 µM. 

A further 24 h incubation increased the percentage of early apoptotic cells at the lowest examined concentration of 2.5 μM. Furthermore, at higher concentrations of **11**, **12** and **13** (5 and 10 µM), a significant increase in cells in the late stage of apoptosis was visible (Figure 5).

These results provide valuable insights into the mechanism of action of compounds **11**, **12** and **13** as potential anti-proliferative agents targeting HeLa cells. The concentration- and time-dependent induction of apoptosis highlighted the effectiveness of these compounds in promoting programmed cell death in the tested cell line.

#### 2.3.3. Caspase Activation

Apoptosis induction was further determined by examining the effects of **11**, **12** and **13** on caspase activation in HeLa cells. Caspase activity induction was determined with the use of the fluorescently labeled caspase inhibitor-FAM-VAD-FMK (a carboxyfluorescein derivative of valylalanylaspartic acid fluoromethyl ketone). The caspase inhibitor binds to active caspases inhibiting their enzymatic activity, thus allowing caspase activity quantification through determining the fluorescent intensity of the bound inhibitor. The results shown in Figure 6 indicated that compounds **11**, **12** and **13** induced caspase activity in HeLa cells in a dose-dependent manner. Increased caspase activation was shown by the increased fluorescence of the caspase inhibitor in the cell population, as indicated in Figure 6. The results showed that compounds **11**, **12** and **13** induced apoptosis through caspase activation in HeLa cells.

By targeting caspase activation, compounds **11**, **12** and **13** initiate the cascade of events leading to programmed cell death. This observation supports the hypothesis that the anti-proliferative effects of these compounds in HeLa cells are mediated through the induction of apoptosis. The evaluation of caspase activity added valuable insight into the mechanism of action of compounds **11**, **12** and **13**, highlighting their potential as apoptotic inducers in cancer therapy.

#### 2.3.4. Cell Cycle Distribution

The effects of compounds **11**, **12** and **13** on the cell cycle phase distribution of HeLa cells were assessed by flow cytometry. The cells were treated with 5, 7 and 10 µM concentrations of **11The r12** and **13** for 48 h and stained with PI. Results presented in Figure 7 showed an increase in the percentage of cells in the sub-G1 phase of the cell cycle upon treatment with compounds **11**, **12** and **13**. Furthermore, the treatment of cells with **12** at the concentrations of 7 and 10 μM also induced G2/M arrest in HeLa cells. These results pointed to the effects of **11**–**13** on DNA fragmentation, which could be indicative of internucleosomal DNA fragmentation induction by **11**–**13**, a hallmark of apoptosis induced by caspase-activated DNase (CAD). 

Taken together, these results indicated that compounds **11**, **12** and **13** exerted their anti-proliferative effects in HeLa cells by inducing DNA fragmentation and potentially triggering apoptosis through the activation of CAD. The cell cycle analysis provided valuable insights into the mechanisms of action of these compounds and their potential as anti-cancer agents.

### 2.4. In Vitro Metabolic Stability Assay

The three most potent compounds (**11**, **12 13**) were tested in an in vitro metabolic stability assay. Human liver microsomes along with NADPH were used to assess their susceptibility to undergo first-phase oxidation reactions. The progress of biotransformation was followed by liquid chromatography–mass spectrometry. The results derived from triplicate incubations, expressed as in vitro metabolic half-life (t_½_), are shown in Figure 8. In order to support the in vitro results, we performed in silico calculations using the Human Liver Microsome-based model for CYP-mediated oxidations provided by the Xenosite online tool [45].

In relation to their stability, the compounds could be ordered as **12** > **13** > **11**, with decreasing stability. All substituents in the R^1^ position can undergo oxidation, and this occurs for substituents in several positions for 4-chlorophenyl (**11**) and 1-naphthyl (**13**). Interestingly, in opposition to the Xenosite’s results, the derivative bearing a piperonyl moiety (**12**) exhibited the best stability in vitro among the studied set of compounds. A detailed survey of possible reasons for this property suggested that another part of the studied molecules can be more important for metabolic stability than the R^1^ substituents, thus diminishing their influence.

The most probable hypothesis includes oxidation of the sulfur atom in the thione functionality, resulting in the formation of sulfenic and, subsequently, sulfinic acid. This kind of biotransformation was reported several times in the literature, including in a detailed study of several thioureas and thiones by Henderson and others [46] and also in a paper by Yamazaki et al. for methimazole [47]. 

## 3. Materials and Methods

### 3.1. Synthesis

The melting points were uncorrected and measured using a Thermogalen (Leica, Vienna, Austia) apparatus. The IR spectra were measured on a Thermo Mattson Satellite FTIR spectrometer (Thermo Mattson, Madison, WI, USA) in KBr pellets; the absorption range was 400–4000 cm^−1^. The ^1^H NMR and ^13^C NMR spectra were recorded on a Varian Gemini 200 apparatus or a Varian Unity Plus 500 apparatus (Varian, Palo Alto, CA, USA), as well as on a Bruker Ascend 600 spectrometer (Bruker, Billerica, MS, USA). The chemical shifts are expressed at δ values relative to Me_4_Si (TMS) as an internal standard. The apparent resonance multiplicity is described as: s (singlet), br s (broad singlet), d (doublet), t (triplet) and m (multiplet). Elemental analyses were performed on a PerkinElmer 2400 Series II CHN Elemental Analyzer (Perkin Elmer, Shelton, CT, USA), and the results indicated by the symbols of the elements were within ±0.4% of the theoretical values. Thin-layer chromatography (TLC) was performed on Merck Kieselgel 60 F254 plates (Merck, Darmstadt, Germany) and visualized by UV spectroscopy. An HPLC-UV analysis was performed on anAgilent 1260 liquid chromatograph equipped with a VWD detector (Agilent, Santa Clara, CA, USA). A Poroshell EC-C18 column (150 × 3 mm, 2.7 um) (Agilent, Santa Clara, CA, USA) was used at the flow rate of 0.2 mL/min. The injection volume was 5 μL. Gradient elution was applied as follows: a linear increase of acetonitrile in water from 5% to 100% over 30 min. Detection was performed at 254 nm.

The commercially unavailable *N*-(2-alkylthio-4-chlorobenzenesulfonyl)cyanamide potassium salts were obtained according to the following methods described previously: **1**, **7** [41], **2**–**3**, **6** [42], **4** [43], **5** [44].

*General procedure for the synthesis of 2-alkythio-4-chloro-N-[imino(heteroaryl)methyl]benzenesulfonamide* **(8–24)**

*Method A.* A mixture of monopotassium salt (1.5 mmol), *p*-toluenesulfonic acid monohydrate (PTSA) (1.5 mmol) and an appropriate thiol (1.5 mmol) in dry toluene (25 mL) was stirred at reflux for 14–28 h. After cooling to room temperature, an insoluble side product was filtered out. The organic layer was washed with water (2 × 10 mL), then dried with MgSO_4_ and concentrated in vacuum. The residue was dissolved in a hot solvent (acetonitrile for compounds **8** and 1**2**, benzene for **14**, ethanol for **15**) and left to crystallize at room temperature. The precipitate was collected by filtration and dried.

*Method B.* A mixture of monopotassium salt (1.5 mmol), PTSA (1.5 mmol) and an appropriate thiol (1.5 mmol) in dry *p*-dioxane (8 mL) was stirred at 105 °C for 4–5 h. After cooling to room temperature, the mixture was concentrated in vacuum to dryness, and the residue was treated with water (20 mL) and stirred using an ultrasonic bath for 5 min. The precipitate was filtered off, dried and crystallized from ethanol (compounds **9** and **16**) or ethanol/acetonitrile mixture (*v*/*v* = 4:1) (compound **13**). 

*Method C.* A mixture of monopotassium salt (1.5 mmol), PTSA (1.5 mmol) and an appropriate thiol (1.5 mmol) in dry toluene (25 mL) was stirred at reflux for 14 h. After cooling to room temperature, the solid was filtered off and dried. The products were purified by crystallization from acetonitrile (compound **19**).

*Method D.* A mixture of monopotassium salt (1.5 mmol), PTSA (1.5 mmol) and an appropriate thiol (1.5 mmol) in dry *p*-dioxane (8 mL) was stirred at 105 °C for 1.5–7 h. After cooling to room temperature, an insoluble side product was filtered out, then the filtrate was concentrated in vacuum to dryness, and the residue was treated with water (20 mL) and stirred using an ultrasonic bath for 5 min. The precipitate was filtered off, dried and crystalized from ethanol (compounds **20**–**24**) or purified by gravity liquid chromatography using silica gel with pore size 60 Å, 220–440 mesh particle size and 35–75 μm particle size (compounds **10**, **11**, **17**, **18**).


**2-Benzylthio-4-chloro-*N*-[imino(3-methyl-2-thioxo-2,3-dihydro-1*H*-imidazol-1-yl)methyl]-5-methylbenzenesulfonamide (8)**


*Method A*. Starting from **1** (0.580 g), 1-methyl-1*H*-imidazole-2-thiol (0.171 g) and PTSA (0.285 g) in toluene for 14 h, the title compound **8** was obtained (0.485 g, 70%): m.p. 163.4–165 °C dec.; HPLC (purity 98.73%): *t*_R_ = 32.5 min.; IR (KBr) v_max_ 3379, 3207, 3169, 3129 (NH), 3003 (CH_Ar_), 1637, 1543 (C=C, C=N); 1378, 1137 (SO_2_) cm^−1^; ^1^H NMR (500 MHz, DMSO-*d_6_*) δ 2.34 (s, 3H, CH_3_), 3.53 (s, 3H, CH_3_-N), 4.36 (s, 2H, CH_2_), 7.22–7.25 (m, 3H, arom.), 7.30–7.31 (m, 2H, arom.), 7.4 (d, 1H, H-4 imidazole), 7.55 (d, 1H, H-5 imidazole), 7.62 (s, 1H, H-3 arom.), 7.95 (s, 1H, H-6 arom.), 8.82 (s, 1H, NH), 11.29 (s, 1H, SO_2_NH) ppm; ^13^C NMR (150.9 MHz, DMSO-*d_6_*) δ 19.43, 35.02, 36.52, 114.89, 120.89, 127.73, 128.76, 128.85, 129.30, 131.15, 132.82, 136.26, 136.76, 137.97, 138.08, 151.32, 162.20 ppm. Found: C, 48.50; H, 4.10; N, 11.88. Calc. for C_19_H_19_ClN_4_O_2_S_3_: C, 48.86; H, 4.10; N, 12.00%.


**4-Chloro-*N*-[imino(3-methyl-2-thioxo-2,3-dihydro-1*H*-imidazol-1-yl)methyl]-5-methyl-2 -(3-trifluoromethylbenzylthio)benzenesulfonamide (9)**


*Method B*. Starting from **2** (0.688 g), 1-methyl-1*H*-imidazole-2-thiol (0.171 g) and PTSA (0.285 g) in *p*-dioxane for 3.5 h, the title compound **9** was obtained (0.401 g, 50%): m.p. 145.7–146.7 °C; HPLC (purity 98.76%): *t*_R_ = 33.05 min; IR (KBr) v_max_ 3398, 3151, 3120 (NH), 2928 (CH _Ar_), 1649, 1548 (C=C, C=N), 1329, 1135 (SO_2_) cm^−1^; ^1^H NMR (500 MHz, DMSO-*d_6_*) δ 2.32 (s, 3H, CH_3_), 3.51 (s, 3H, CH_3_-N), 4.48 (s, 2H, CH_2_), 7.36 (d, 1H, H-4 imidazole), 7.44 (t, 1H, arom.), 7.53–7.62 (m, 4H, H-5 imidazole and arom.), 7.70 (s, 1H, arom.), 7.95 (s, 1H, H-6 arom.), 8.83 (s, 1H, NH), 11.29 (s, 1H, NHSO_2_) ppm; ^13^C NMR (125 MHz, DMSO-*d_6_*) δ 19.64, 35.22, 35.91, 115.10, 121.11, 124.67, 126.20, 129.25, 129.56, 129.80, 130.14, 131.48, 133.38, 133.54, 135.61, 138.31, 138.40, 138.84, 151.50, 162.44 ppm. Found: C, 45.25; H, 3.50; N, 10.76. Calc. for C_20_H_18_ClF_3_N_4_O_2_S_3_: C, 44.90; H, 3.39; N, 10.47%. 


**4-Chloro-*N*-[imino(3-methyl-2-thioxo-2,3-dihydro-1*H*-imidazol-1-yl)methyl]-5-methyl-2 -(4-trifluoromethylbenzylthio)benzenesulfonamide (10)**


*Method D*. Starting from **3** (0.688 g), 1-methyl-1*H*-imidazole-2-thiol (0.171 g) and PTSA (0.285 g) in *p*-dioxane for 3 h, after purification on silica gel using CHCl_3_/MeOH (*v*/*v* = 8:1) as the eluent, the title compound **10** was obtained (0.369 g, 46%): m.p. 183.3–184.3 °C; HPLC (purity 99.76%): *t*_R_ = 33.15 min; IR (KBr) v_max_ 3385, 3201 (NH), 3033 (CH_Ar_), 1648, 1534 (C=N, C=C), 1320, 1165 (SO_2_) cm^−1^; ^1^H NMR (500 MHz, DMSO-*d_6_*) δ 2.33 (s, 3H, CH_3_), 3.51 (s, 3H, CH_3_-N), 4.47 (s, 2H, CH_2_), 7.38 (d, 1H, H-4 imidazole), 7.51 (d, 2H, arom.), 7.55 (d, 1H, H-5 imidazole), 7.56 (d, 2H, arom.), 7.62 (s, 1H, arom.), 7.95 (s, 1H, arom.), 8.80 (s, 1H, NH), 11.28 (s, 1H, NHSO_2_) ppm; ^13^C NMR (150.9 MHz, DMSO-*d_6_*) δ 19.44, 34.99, 35.79, 114.98, 120.91, 125.63, 125.65, 125.68, 128.92, 131.28, 133.21, 135.49, 138.16, 138.19, 142.13, 151.33, 162.20 ppm. Found: C, 45.40; H, 3.56; N, 11.00. Calc. for C_20_H_18_ClF_3_N_4_O_2_S_3_: C, 45.51; H, 3.62; N, 11.17%.


**4-Chloro-*N*-[imino(3-methyl-2-thioxo-2,3-dihydro-1*H*-imidazol-1-yl)methyl]-5-methyl-2-(4-chlorobenzylthio)benzenesulfonamide (11)**


*Method D*. Starting from **4** (0.638 g), 1-methyl-1*H*-imidazole-2-thiol (0.171 g) and PTSA (0.285 g) in *p*-dioxane for 4 h, after purification on silica gel using CHCl_3_/MeOH (*v*/*v* = 8:1) as the eluent, the title compound **11** was obtained (0.286 g, 38%): m.p. 176.3–177.8 °C; HPLC (purity 99.69%): *t*_R_ = 33.25 min; IR (KBr) v_max_ 3355, 3201(NH), 3020 (CH_Ar_), 1633, 1542 (C=N, C=C), 1346, 1138 (SO_2_) cm^−1^; ^1^H NMR (500 MHz, DMSO-*d_6_*) δ 2.33 (s, 3H, CH_3_), 3.52 (s, 3H, CH_3_-N), 4.36 (s, 2H, CH_2_), 7.25 (d, *J* = 8.3 Hz, 2H, 4-Cl-Ph arom.), 7.30 (d, *J* = 8.3 Hz, 2H, 4-Cl-Ph arom.), 7.38 (d, 1H, H-4 imidazole), 7.52 (d, 1H, H-5 imidazole), 7.61 (s, 1H, H-3 arom.), 7.94 (s, 1H, H-6 arom.), 8.79 (s, 1H, NH), 11.28 (s, 1H, NHSO_2_) ppm; ^13^C NMR (150.9 MHz, DMSO-*d_6_*) δ 19.45, 35.04, 35.67, 114.97, 120.90, 128.80, 128.93, 131.04, 131.21, 132.29, 133.06, 135.80, 136.13, 138.12, 138.16, 151.32, 162.18 ppm. Found: C, 45.46; H, 3.60; N, 11.15. Calc. for C_19_H_18_Cl_2_N_4_O_2_S_3_: C, 45.51; H, 3.62; N, 11.17%.

**4-Chloro-2-(6-chlorobenzo** [1,3]**dioxol-5-ylmethylthio)-2-*N*-[imino(3-methyl-2-thioxo-2,3-dihydro-1*H*-imidazol-1-ylmethyl]-5-methylbenzenesulfonamide (12)**

*Method A*. Starting from **5** (0.704 g), 1-methyl-1*H*-imidazole-2-thiol (0.171 g) and PTSA (0.285 g) in toluene for 15 h, the title compound **12** was obtained (0.381 g, 47%): m.p. 177–180 °C; HPLC (purity 94.26%): *t*_R_ = 33.38 min; IR (KBr) v_max_ 3341, 3188, 3155, 3133 (NH), 3044 (CH_Ar_), 1644, 1533 (C=N, C=C), 1343, 1127 (SO_2_) cm^−1^; ^1^H NMR (500 MHz, DMSO-*d_6_*) δ 2.36 (s, 3H, CH_3_), 3.50 (s, 3H, CH_3_-N), 4.29 (s, 2H, CH_2_), 6.04 (s, 2H, O-CH_2_-O), 6.93 (s, 1H, arom.), 6.99 (s, 1H, arom.), 7.33 (d, *J* = 2.7 Hz, 1H, H-4 imidazole), 7.50 (d, *J* = 2.7 Hz, 1H, H-5 imidazole), 7.60 (s, 1H, H-3 arom.), 7.95 (s, 1H, H-6 arom.), 8.77 (s, 1H, NH), 11.27 (s, 1H, SO_2_NH) ppm; ^13^C NMR (150.9 MHz, DMSO-*d_6_*) δ 19.47, 33.99, 35.10, 102.60, 110.07, 110.59, 114.88, 120.81, 125.63, 127.12, 128.98, 131.19, 133.16, 136.15, 138.18, 138.21, 147.03, 148.08, 151,38, 162.23 ppm. Found: C, 44.00; H, 3.20; N, 9.99. Calc. For C_20_H_18_Cl_2_N_4_O_4_S_3_: C, 44.04; H, 3.33; N, 10.27%.


**4-Chloro-*N*-[imino(3-methyl-2-thioxo-2,3-dihydro-1*H*-imidazol-1-yl)methyl]-5-methyl- 2-(naphthalen-1-ylmethylthio)benzenesulfonamide (13)**


*Method B*. Starting from **6** (0.661 g), 1-methyl-1*H*-imidazole-2-thiol (0.171 g) and PTSA (0.285 g) in *p*-dioxane for 4 h, the title compound **13** was obtained (0.396 g, 51%): m.p. 197–198 °C; HPLC (purity 98.23%): *t*_R_ = 33.96 min; IR (KBr) v_max_ 3349, 3201, 3164, 3123 (NH), 2996 (CH_Ar_),1637, 1533 (C=C, C=N), 1386, 1133 (SO_2_); ^1^H NMR (500 MHz, DMSO-*d_6_*) δ 2.36 (s, 3H, CH_3_), 3.49 (s, 3H, CH_3_-N), 4.82 (s, 2H, CH_2_), 7.28 (d, 1H, H-4 imidazole), 7.38 (d, 1H, H-5 imidazole), 7.40 (d, 1H, arom.), 7.45–7.56 (m, 3H, arom.), 7.73 (s, 1H, H-3 arom), 7.85 (d, 1H, arom.), 7.92 (d, 1H, arom.), 7.97 (s, 1H, H-6 arom.), 8.14 (d, 1H, arom.), 8.76 (s, 1H, NH), 11.18 (s, 1H, NHSO_2_) ppm; ^13^C NMR (125 MHz, DMSO-*d_6_*) δ 19.70, 35.10, 35.22, 114.96, 120.97, 124.65, 126.09, 126.60, 126.80, 128.47, 128.98, 129.08, 129.24, 131.30, 131.94, 132.25, 133.03, 134.11, 137.15, 137.96, 138.43, 151.42, 162.34 ppm. Found: C, 53.62; H, 4.21; N, 11.11. Calc. for C_23_H_21_ClN_4_O_2_S_3_: C, 53.42; H, 4.09; N, 10.84%. 


**4-Chloro-2-ethoxycarbonylmethylthio-*N*-[imino(3-methyl-2-thioxo-2,3-dihydro-1*H*-imidazol-1-yl)methyl]-5-methylbenzenesulfonamide (14)**


*Method A*. Starting from **7** (0.508 g), 1-methyl-1*H*-imidazole-2-thiol (0.171 g) and PTSA (0.285 g) in toluene for 15 h, the title compound **14** was obtained (0.076 g, 11%): m.p. 125–126 °C; HPLC (purity 97.93%): *t*_R_ = 33.06 min; IR (KBr) v_max_ 3341, 3208, 3129 (NH), 2960 (CH_3_, CH_2_), 1722 (C=O), 1647, 1627, 1560 (C=N, C=C), 1345, 1145 (SO_2_) cm^−1^; ^1^H NMR (500 MHz, DMSO-*d_6_*) δ 1.10 (t, 3H, CH_3_), 2.35 (s, 3H, CH_3_), 3.48 (s, 3H, CH_3_-N), 4.01–4.05 (m, 4H, O-CH_2_, S-CH_2_), 7.35 (d, 1H, H-4 imidazole), 7.52 (s, 1H, H-3 arom.), 7.59 (d, 1H, H-5 imidazole), 7.96 (s, 1H, H-6), 8.84 (s, 1H, NH), 11.27 (s, 1H, SO_2_NH) ppm; ^13^C NMR (150.9 MHz, DMSO-*d_6_*) δ 14.39, 19.43, 34.63, 34.99, 61.56, 115.10, 120.85, 128.54, 131.21, 133.22, 135.38, 137.95, 138.16, 151.23, 162.21, 169.23 ppm. Found: C, 41.45; H, 4.09; N, 12.02. Calc. for C_16_H_19_ClN_4_O_4_S_3_: C, 41.51; H, 4.14; N, 12.10%.


**2-Benzylthio-4-chloro-5-methyl-*N*-[imino(4-methyl-5-thioxo-4,5-dihydro-1*H*-1,2,4-triazol-1-yl)methyl]benzenesulfonamide (15)**


*Method A*. Starting from **1** (0.580 g), 4-methyl-4*H*-1,2,4-triazole-3-thiol (0.173 g) and PTSA (0.285 g) in toluene for 28 h, the title compound **15** was obtained (0.239 g, 34%): m.p. 167–170 °C; HPLC (purity 94.05%): *t*_R_ = 31.42 min; IR (KBr) v_max_ 3383 (NH), 3087, 3064 (CH_Ar_), 2921 (CH_3_, CH_2_), 1650, 1557 (C=N, C=C), 1356, 1138 (SO_2_) cm^−1^; ^1^H NMR (200 MHz, DMSO-*d_6_*) δ 2.36 (s, 3H, CH_3_), 3.50 (s, 3H, CH_3_-N), 4.32 (s, 2H, CH_2_), 7.19–7.31 (m, 5H, arom.), 7.58 (s, 1H, H-3 arom.), 7.97 (s, 1H, H-6 arom.), 8.74 (s, 2H, NH and H-5 triazole), 10.25 (s, 1H, SO_2_NH) ppm; ^13^C NMR (125 MHz, DMSO-*d_6_*) δ 19.41, 32.91, 36.89, 127.64, 128.79, 129.34, 131.06, 133.08, 136.21, 136.69, 137.94, 138.50, 143.37, 151.03, 167.79 ppm. Found: C, 45.98; H, 3.62; N, 14.73. Calc. for C_18_H_18_ClN_5_O_2_S_3_: C, 46.19; H, 3.88; N, 14.96%. 


**4-Chloro-*N*-[imin(4-methyl-5-thioxo-4,5-dihydro-1*H*-1,2,4-triazol-1-yl)methyl]-5-methyl-2-(3-trifluoromethylbenzylthio)benzenesulfonamide (16)**


*Method B*. Starting from **2** (0.688 g), 4-methyl-4*H*-1,2,4-triazole-3-thiol (0.173 g) and PTSA (0.285 g) in *p*-dioxane for 5 h, the title compound **16** was obtained (0.386 g, 48%): m.p. 170–173 °C; HPLC (purity 97.51%): *t*_R_ = 32.15 min; IR (KBr) v_max_ 3363 (NH), 2924, 2853 (CH_3_, CH_2_), 1649, 1542 (C=N, C=C), 1331, 1134 (SO_2_) cm^−1^; ^1^H NMR (500 MHz, DMSO-*d_6_*) δ 2.32 (s, 1H, CH_3_), 3.48 (s, 3H, CH_3_-N), 4.44 (s, 2H, CH_2_), 7.44 (t, 1H, arom.), 7.54–7.59 (m, 2H, arom.), 7.61 (d, 1H, arom.), 7.68 (s, 1H, arom.), 7.98 (s, 1H, H-6 arom.), 8.73 (s, 1H, H-5 triazole), 8.76 (s, 1H, NH), 10.22 (s, 1H, SO_2_NH) ppm; ^13^C NMR (150.9 MHz, DMSO-*d_6_*) δ 19.41, 32.89, 36.17, 123.63, 124.38, 124.41, 125.43, 126.05, 126.08, 129.84, 129.91, 131.19, 133.44, 133.53, 135.26, 137.94, 138.58, 138.89, 143.41, 150.99, 167.83 ppm. Found: C, 42.46; H, 3.16; N, 13.01. Calc. for C_19_H_17_ClF_3_N_5_O_2_S_3_: C, 42.57; H, 3.20; N, 13.07%. 


**4-Chloro-*N*-[imino(4-methyl-5-thioxo-4,5-dihydro-1*H*-1,2,4-triazol-1-yl)methyl]-5-methyl-2-(4-trifluoromethylbenzylthio)benzenesulfonamide (17)**


*Method D*. Starting from **3** (0.688 g), 4-methyl-4*H*-1,2,4-triazole-3-thiol (0.173 g) and PTSA (0.285 g) in *p*-dioxane for 7 h, after purification on silica gel using CHCl_3_/MeOH (*v*/*v* = 8:1) as the eluent, the title compound **17** was obtained (0.338 g, 42%): m.p. 188.8–190.3 °C; HPLC (purity 98.65%): *t*_R_ = 32.31 min; IR (KBr) v_max_ 3380, (NH), 3088, 3066 (CH_Ar_), 2924, 2854 (CH_3_, CH_2_), 1654, 1559 (C=N, C=C), 1344, 1137 (SO_2_) cm^−1^; ^1^H NMR (500 MHz, DMSO-*d_6_*) δ 2.33 (s, 1H, CH_3_), 3.49 (s, 3H, CH_3_-N), 4.44 (s, 2H, CH_2_), 7.53 (d, 2H, arom.), 7.57 (d, 3H, arom.), 7.97 (s, 1H, H-6 arom.), 8.75 (s, 2H, NH and H-5 triazole), 10.22 (s, 1H, SO_2_NH) ppm; ^13^C NMR (150.9 MHz, DMSO-*d_6_*) δ 19.43, 32.91, 36.22, 125.59, 125.61, 125.64, 129.61, 131.14, 131.19, 133.50, 135.42, 138.01, 138.81, 142.05, 143.43, 151.03, 167.83 ppm. Found: C, 42.41; H, 3.13; N, 13.05. Calc. for C_19_H_17_ClF_3_N_5_O_2_S_3_: C, 42.57; H, 3.20; N, 13.07%.


**4-Chloro-2-(4-chlorobenzylthio)-*N*-[imino(4-methyl-5-thioxo-4,5-dihydro-1*H*-1,2,4-triazol-1-yl)methyl]5-methylbenzenesulfonamide (18)**


*Method D*. Starting from **4** (0.638 g), 4-methyl-4*H*-1,2,4-triazole-3-thiol (0.173 g) and PTSA (0.285 g) in *p*-dioxane for 1.5 h, after purification on silica gel using CHCl_3_/MeOH (*v*/*v* = 8:1) as the eluent, the title compound **18** was obtained (0.249 g, 33%): m.p. 184.8–185.7 °C; HPLC (purity 98.75%): *t*_R_ = 32.33 min; IR (KBr) v_max_ 3379, (NH), 3085, 3066 (CH_Ar_), 2924, 2854 (CH_3_, CH_2_), 1652, 1557 (C=N, C=C), 1345, 1137 (SO_2_) cm^−1^; ^1^H NMR (500 MHz, DMSO-*d_6_*) δ 2.33 (s, 1H, CH_3_), 3.49 (s, 3H, CH_3_-N), 4.33 (s, 2H, CH_2_), 7.26 (d, *J* = 8.3 Hz, 2H, arom.), 7.32 (d, *J* = 8.3 Hz, 2H, arom.), 7.57 (s, 1H, H-3 arom.), 7.96 (s, 1H, H-6 arom.), 8.73 (s, 1H, NH), 8.75 (s, 1H, H-5 triazole), 10.22 (s, 1H, SO_2_NH) ppm; ^13^C NMR (150.9 MHz, DMSO-*d_6_*) δ 19.43, 32.94, 36.04, 128.74, 129.53, 131.13, 132.20, 133.31, 135.74, 136.04, 137.97, 138.73, 143.40, 151.02, 167.81 ppm. Found: C, 42.89; H, 3.35; N, 13.88. Calc. for C_19_H_17_ClF_3_N_5_O_2_S_3_: C, 43.03; H, 3.41; N, 13.94%. 


**2-Benzylthio-4-chloro-5-methyl-*N*-[imino-(2-thioxo-2,3-dihydro-1*H*-benzo[d]imidazol-1-yl)methyl]benzenesulfonamide (19)**


*Method C*. Starting from **1** (0.580 g), 1*H*-benzo[*d*]imidazole-2-thiol (0.225 g) and PTSA (0.285 g) in toluene for 14 h, the title compound **19** was obtained (0.228 g, 30%): m.p. 182–185 °C; HPLC (purity 91.23%): *t*_R_ = 33.69 min; IR (KBr) v_max_ 3343 (NH), 3053 (CH_Ar_), 1630, 1546 (C=N, C=C), 1342, 1138 (SO_2_) cm^−1^; ^1^H NMR (500 MHz, DMSO-*d_6_*) δ 2.35 (s, 3H, CH_3_), 4.34 (s, 2H, CH_2_), 7.12–7.18 (m, 2H, arom.), 7.24–7.33 (m, 6H, arom.), 7.59 (s, 1H, H-3 arom.), 7.89 (d, 1H, arom.), 8.04 (s, 1H, H-6 arom.), 8.82 (s, 1H, NH), 10.50 (s, 1H, SO_2_NH), 13.65 (s, 1H, NH benzimidazole) ppm; ^13^C NMR (125 MHz, DMSO-*d_6_*) δ 19.41, 36.59, 110.49, 115.62, 122.76, 123.71, 125.57, 127.72, 128.31, 128.76, 129.39, 131.10, 131.57, 132.74, 136.35, 136.72, 137.27, 138.30, 152.46, 168.78 ppm. Found: C, 52.90; H, 4.10; N, 11.52. Calc. for C_22_H_19_ClN_4_O_2_S_3_: C, 52.53; H, 3.81; N, 11.14%.


**4-Chloro-*N*-[imino-(2-thioxo-2,3-dihydro-1*H*-benzo[d]imidazol-1-yl)methyl]-5-methyl-2-(3-trifluoromethylbenzylthio)benzenesulfonamide (20)**


*Method D*. Starting from **2** (0.688 g), 1*H*-benzo[*d*]imidazole-2-thiol (0.225 g) and PTSA (0.285 g) in *p*-dioxane for 6 h, the title compound **20** was obtained (0.428 g, 50%): m.p. 168–172 °C; HPLC (purity 94.55%): *t*_R_ = 34.05 min; IR (KBr) v_max_ 3365, 3228 (NH), 2932 (CH), 1633, 1545 (C=N, C=C), 1329, 1129 (SO_2_) cm^−1^; ^1^H NMR (500 MHz, DMSO-*d_6_*) δ 2.33 (s, 3H, CH_3_), 4.56 (s, 2H, CH_2_), 7.11 (t, 1H, arom.), 7.20–7.28 (m, 2H, arom.), 7.35 (t, 1H, arom.), 7.75 (d, 1H, arom.), 7.56 (d, 1H, arom.), 7.58 (s, 1H, H-3 arom.), 7.63 (s, 1H, arom.), 7.86 (s, 1H, arom.), 8.03 (s, 1H, H-6 arom.), 8.83 (s, 1H, NH), 10.45 (s, 1H, SO_2_NH), 13.63 (s, 1H, NH benzimidazole) ppm; ^13^C NMR (125 MHz, DMSO-*d_6_*) δ 19.64, 36.09, 110.74,115.70, 123.85, 124.69, 125.75, 126.26, 128.96, 129.60, 130.04, 131.35, 131.89, 133.38, 133.70, 136.07, 137.83, 138.47, 138.54, 152.66, 169.02 ppm. Found: C, 48.08; H, 3.13; N, 9.64. Calc. for C_23_H_18_ClF_3_N_4_O_2_S_3_: C, 48.37; H, 3.18; N, 9.81%.


**4-Chloro-*N*-[imino-(2-thioxo-2,3-dihydro-1*H*-benzo[d]imidazol-1-yl)methyl]-5-methyl-2-(4-trifluoromethylbenzylthio)benzenesulfonamide (21)**


*Method D*. Starting from **3** (0.688 g), 1*H*-benzo[*d*]imidazole-2-thiol (0.225 g) and PTSA (0.285 g) in *p*-dioxane for 4 h, the title compound **21** was obtained (0.428 g, 50%): m.p. 188–189 °C; HPLC (purity 95.68%): *t*_R_ = 34.08 min; IR (KBr) v_max_ 3363, (NH), 3027 (CH_Ar_), 2860 (CH_3_, CH_2_), 1648, 1585 (C=N, C=C), 1323, 1155 (SO_2_) cm^−1^; ^1^H NMR (500 MHz, DMSO-*d_6_*) δ 2.33 (s, 3H, CH_3_), 4.45 (s, 2H, CH_2_), 7.10 (t, 1H, arom.), 7.23 (d, 1H, arom.), 7.28 (t, 1H, arom.), 7.44 (d, 2H, arom.), 7.46 (d, 2H, arom.), 7.60 (s,1H, H-3 arom.), 7.83 (d, 1H, arom.), 8.02 (s, 1H, H-6 arom.), 8.83 (s, 1H, NH), 10.50 (s, 1H, SO_2_NH), 13.65 (s, 1H, NH benzimidazole) ppm; ^13^C NMR (125 MHz, DMSO-*d_6_*) δ 19.66, 36.06, 110.74, 115.79, 123.92, 125.78, 128.84, 130.29, 131.33, 131.83, 133.40, 136.05, 137.91, 138.57, 142.01, 152.70, 169.01 ppm. Found: C, 47.99; H, 3.05; N, 9.53. Calc. for C_23_H_18_ClF_3_N_4_O_2_S_3_: C, 48.37; H, 3.18; N, 9.81%.


**4-Chloro-2-(4-chlorobenzylthio)-*N*-[imino-(2-thioxo-2,3-dihydro-1*H*-benzo[d]imidazol-1-yl)methyl]-5-methylbenzenesulfonamide (22)**


*Method D*. Starting from **4** (0. 638 g), 1*H*-benzo[*d*]imidazole-2-thiol (0.225 g) and PTSA (0.285 g) in *p*-dioxane for 4.5 h, the title compound **22** was obtained (0.322 g, 40%): m.p. 181.8–183.5 °C; HPLC (purity 94.42%): *t*_R_ = 34.28 min; IR (KBr) v_max_ 3386, 3186 (NH), 3081, 3038 (CH_Ar_), 2922, 2856 (CH_3_, CH_2_), 1639, 1543 (C=N, C=C), 1345, 1137 (SO_2_) cm^−1^; ^1^H NMR (500 MHz, DMSO-*d_6_*) δ 2.33 (s, 3H, CH_3_), 4.33 (s, 2H, CH_2_), 7.08–7.18 (m, 3H, arom.), 7.22–7.27 (m, 3H, arom.), 7.29 (t, 1H, arom.), 7.58 (s,1H, H-3 arom.), 7.84 (d, 1H, arom.), 8.02 (s, 1H, H-6 arom.), 8.81 (s, 1H, NH), 10.50 (s, 1H, SO_2_NH), 13.65 (s, 1H, NH benzimidazole) ppm; ^13^C NMR (125 MHz, DMSO-*d_6_*) δ 19.66, 35.94, 110.75, 115.82, 123.94, 125.82, 128.79, 128.91, 131.34, 131.81, 132.49, 133.23, 135.97, 136.40, 137.79, 138.54, 152.68, 169.00 ppm. Found: C, 48.78; H, 3.36; N, 10.03. Calc. for C_22_H_18_Cl_2_N_4_O_2_S_3_: C, 49.19; H, 3.38; N, 10.42%.


**4-Chloro-2-(6-chlorobenzo [1,3]dioxol-5-ylmethylthio)-*N*-[imino-(2-thioxo-2,3-dihydro-1*H*-benzo[d]imidazol-1-yl)methyl]-5-methylbenzenesulfonamide (23)**


*Method D*. Starting from **5** (0. 638 g), 1*H*-benzo[*d*]imidazole-2-thiol (0.225 g) and PTSA (0.285 g) in *p*-dioxane for 6 h, the title compound **23** was obtained (0.358 g, 41%): m.p. 213–214 °C; HPLC (purity 92.21%): *t*_R_ = 34.16 min; IR (KBr) v_max_ 3367, 3262 (NH), 2975, 2847 (CH), 1627, 1505 (C=N, C=C), 1342, 1137 (SO_2_) cm^−1^; ^1^H NMR (500 MHz, DMSO-*d_6_*) δ 2.36 (s, 3H, CH_3_), 4.26 (s, 2H, CH_2_), 6.00 (s, 2H, CH_2_-O), 6.84 (d, 2H, arom.), 7.05 (t, 1H, arom.), 7.18–7.28 (m, 2H, arom.), 7.59 (s,1H, H-3 arom.), 7.77 (d, 1H, arom.), 8.04 (s, 1H, H-6 arom.), 8.78 (s, 1H, NH), 10.56 (s, 1H, SO_2_NH), 13.63 (s, 1H, NH benzimidazole) ppm; ^13^C NMR (125 MHz, DMSO-*d_6_*) δ 19.70, 35.35, 67.04, 102.76, 110.06, 110.66, 110.88, 115.98, 123.83, 125.76, 127.03, 128.99, 131.28, 131.81, 133.38, 136.61, 137.93, 138.59, 147.15, 148.22, 152.81, 169.00 ppm. Found: C, 47.39; H, 3.17; N, 9.30. Calc. for C_23_H_18_Cl_2_N_4_O_4_S_3_: C, 47.50; H, 3.12; N, 9.63%.


**4-Chloro-*N*-[imino-(2-thioxo-2,3-dihydro-1*H*-benzo[*d*]imidazol-1-yl)methyl]-5-methyl-2-(naphthalen-1-ylmethylthio)benzenesulfonamide (24)**


*Method D*. Starting from **6** (0. 638 g), 1*H*-benzo[*d*]imidazole-2-thiol (0.225 g) and PTSA (0.285 g) in *p*-dioxane for 6.5 h, the title compound **24** was obtained (0.456 g, 55%): m.p. 192–193 °C; HPLC (purity 92.14%): *t*_R_ = 34.77 min; IR (KBr) v_max_ 3479, 3363 (NH), 3053 (CH_Ar_), 1630, 1546 (C=N, C=C), 1342, 1138 (SO_2_) cm^−1^; ^1^H NMR (500 MHz, DMSO-*d_6_*) δ 2.37 (s, 3H, CH_3_), 4.79 (s, 2H, CH_2_), 6.91 (d, 1H, arom.), 7.18–7.26 (m, 2H, arom.), 7.30–7.36 (m, 2H, arom.), 7.42 (d, 1H, arom.), 7.45 (d, 1H, arom.), 7.67 (d, 1H, arom.), 7.74 (s,1H, H-3 arom.), 7.80 (d, 1H, arom.), 7.86 (d, 1H, arom.), 8.06 (s, 1H, H-6 arom.), 8.09 (d, 1H, arom.), 8.78 (s, 1H, NH), 10.40 (s, 1H, SO_2_NH), 13.58 (s, 1H, NH benzimidazole) ppm; ^13^C NMR (125 MHz, DMSO-*d_6_*) δ 19.70, 35.13, 110.66, 115.53, 123.84, 124.69, 125.67, 126.00, 126.54, 126.73, 128.53, 128.96, 129.05, 129.13, 131.25, 131.27, 131.75, 131.91, 132.09, 133.14, 134.01, 137.33, 137.57, 138.65, 152.59, 168.92 ppm. Found: C, 56.29; H, 4.00; N, 9.90. Calc. for C_26_H_21_ClN_4_O_2_S_3_: C, 56.46; H, 3.83; N, 10.13%.

### 3.2. X-ray Structure Determination

The diffraction intensity data were collected on an IPDS 2T dual-beam diffractometer (STOE & Cie GmbH, Darmstadt, Germany) at 120.0(2) K with CuKα radiation from a microfocus X-ray source (GeniX 3D Cu High Flux, Xenocs, Sassenage, 50 kV, 0.6 mA, λ = 1.54186 Å). The crystal was thermostated in a nitrogen stream at 120 K using the CryoStream-800 device (Oxford CryoSystem, Long Hanborough, UK) during the entire experiment.

Data collection and data reduction were controlled by the X-Area 1.75 program (STOE, 2015). An absorption correction was performed on the integrated reflections by a combination of frame scaling, reflection scaling and a spherical absorption correction. Outliers were rejected according to the Blessing’s method. The structures were solved by direct methods and refined anisotropically using the program packages OLEX2 and SHELX-2015. The positions of the C-H hydrogen atoms were calculated geometrically and taken into account with isotropic temperature factors. The amine hydrogen atoms H2a and H2b were found in the Fourier residual electron density map and were refined with the N–H distance restrained to 0.88(2) Å. 

Computer programs: *X-AREA* WinXpose 2.0.22.0 (STOE, 2016), *X-AREA* Recipe 1.33.0.0 (STOE, 2015), STOE *X-AREA*, ShelXT [48], *SHELXL* [49], Olex2 [50].

The crystal data, data collection and structure refinement details are summarized in Appendix A.

Crystallographic data for the structure of **10** reported in this paper were deposited in the Cambridge Crystallographic Data Centre as a supplementary publication, No. CCDC 1832901. Copies of the data can be obtained free of charge on application to CCDC, 12 Union Road, Cambridge CB2 1EZ, UK (Fax: (þ44) 1223-336-033; Email: deposit@ccdc.cam.ac.uk).

### 3.3. Cell Culture and Cell Viability Assay

All chemicals, if not stated otherwise, were obtained from Sigma-Aldrich (St. Louis, MO, USA). The MCF-7, HeLa and HaCaT cell lines were purchased from Cell Lines Services (Eppelheim, Germany), and the HCT-116 cell line was purchased from ATCC (ATCC-No: CCL-247). The cells were cultured in Dulbecco’s modified Eagle’s medium (DMEM) supplemented with 10% fetal bovine serum, 2 mM glutamine, 100 units/mL of penicillin and 100 μg/mL of streptomycin. The cultures were maintained in a humidified atmosphere with 5% CO_2_ at 37 °C, in an incubator (Heraeus, HeraCell). 

**Cytotoxicity assay:** Cell viability was determined using the MTT (3-(4,5-dimethylthiazol-2-yl)-2,5diphenyltetrazoliumbromide) assay. The cells were seeded in 96-well plates at a density of 5 × 10^3^ cells/well and treated for 24, 48 and 72 h with the examined compounds in the concentration range 1–100 μM. Following treatment, MTT (0.5 mg/mL) was added to the medium, and the cells were further incubated for 2 h at 37 °C. the cells were lysed with DMSO, and the absorbance of the formazan solution was measured at 550 nm with a plate reader (Victor, 1420 multilabel counter). The experiment was performed in triplicate. The values are expressed as the mean ± SD of at least three independent experiments.

**Detection of apoptosis by Annexin V-PE and 7-AAD staining:** Apoptosis induction was detected with an Annexin V-PE Apoptosis Detection Kit I (BD Biosciences, Belgium) according to the manufacturer’s instructions. The cells were treated with **11**–**13** (2.5, 5 and 10 µM) for 24 and 48 h. Following treatment, the cells were collected and stained with Annexin V-phycoerythrin (PE) and 7-amino-actinomycin (7-AAD) in Annexin-binding buffer for 15 min at RT in the dark. The acquisition was performed on a FACSCalibur cytometer (BD), and the data were analyzed with Flowing software (version 2.5).

**Caspase Activity Determination:** Caspase activity was determined with the FLICA Apoptosis Detection Kit (Immunochemistry Technologies) according to the manufacturer’s instructions. The cells were treated with comp. **11**, **12** and **13** (5, 7 and 10 µM) for 24 h, after which the cells were collected and suspended in a buffer containing the caspase inhibitor, i.e., a carboxyfluorescein-labeled fluoromethyl ketone peptide. The cells were subsequently incubated for 1 h at 37 °C under 5% CO_2_ and, next, they were washed with a washing buffer. The fluorescence intensity of fluorescein was determined by flow cytometry (BD FACSCalibur), and caspase activity was determined as the amount of fluorescence emitted from the caspase inhibitors bound to the caspases. The data were analyzed with Flowing software (version 2.5).

**Cell Cycle Distribution Analysis:** The effects of **11**, **12** and **13** on the cell cycle distribution in HeLa cells were determined by flow cytometry analysis. The cells were treated with **11**, **12** and **13** (5, 7 and 10 µM) for 48 h, after which they were fixed in cold 70% ethanol for 24 h. The fixed cells were treated with 100 µg/mL of RNAse (Invitrogen, Germany) and stained with 10 µg/mL of PI (Invitrogen, Germany) for 30 min at RT. The acquisition was performed on a FACSCalibur cytometer (BD), and the data were analyzed with Flowing software (version 2.5).

**Statistical Analysis:** Values are expressed as means ± SD of at least three independent experiments. Statistical analysis was performed using GraphPad Prism 5.0 (GraphPad software). Differences between control and treated samples were analyzed by one-way ANOVA with Tukey’s post hoc tests. A *p*-value < 0.05 was considered statistically significant in each experiment.

### 3.4. In Vitro Metabolic Stability Assay

Stock solutions of the studied compounds were prepared at a concentration of 10 mM in DMSO. Working solutions were prepared by dilution of the stock solutions with reaction buffer or acetonitrile; the final concentration of the organic solvent did not exceed 1%. The incubation mixture contained the studied derivative at a 10 μM concentration, 1 mM NADPH (Sigma-Aldrich) and 0.5 mg/mL of human liver microsomes (HLM, Sigma-Aldrich) in potassium phosphate buffer (0.1 M, pH 7.4). The incubation was carried out in a thermostat at 37 °C and started by the addition of the compound of interest. Then, 20 μL samples were taken after 5, 15, 30, 45 and 60 min. The enzymatic reaction was terminated by the addition of an equal volume of ice-cold acetonitrile containing 10 μM sulfamethoxazole as an internal standard. Control incubations were performed without NADPH as a negative control reflecting NADPH-independent processes, such as chemical degradation and precipitation. After collection, the samples were immediately centrifuged (10 min, 10,000 rpm), and the resulting supernatant was directly analyzed or kept at −80 °C until LC-MS analysis. The natural logarithm of a compound over the IS peak area ratio was plotted vs. the incubation time. The metabolic half-time (t_½_) was calculated from the slope of the linear regression, as demonstrated [51].

The LC-MS analysis was performed on an Agilent 1260 system coupled to SingleQuad 6120 mass spectrometer (Agilent Technologies, Santa Clara, CA, USA). Poroshell EC-C18 (2.1 mm × 150 mm, 2.7 μm, Agilent Technologies, Santa Clara, CA, USA) was used in reversed-phase mode with gradient elution starting with 90% of phase A (0.1% formic acid in water) and 10% of phase B (0.1% dormic acid in acetonitrile). The amount of phase B was linearly increased to 100% in 15 min and equilibrated. The total analysis time was 21 min at 25 °C, the flow rate was 0.5 mL/min, and the injection volume was 20 μL. The mass spectrometer was equipped with an electrospray ion source and operated in positive ionization mode. The mass analyzer was set for each individual compound to detectthe [M+H]^+^ protonated molecule. The MSD parameters of the ESI source were as follows: nebulizer pressure 40 psig (N_2_), drying gas 10 mL/min (N_2_), drying gas temperature 300 °C, capillary voltage 3.0 kV, fragmentor voltage 150 V.

## 4. Conclusions

In conclusion, our study focused on the design and synthesis of novel benzenesulfonamide derivatives containing imidazole rings as potential anticancer agents. The cytotoxic evaluation revealed that compounds **11**–**13** exhibited remarkable selectivity and efficacy against HeLa cervical cancer cells, with IC_50_ values of 6–7 μM. These compounds showed significantly lower cytotoxicity towards the non-tumor cell line HaCaT (IC_50_: 18–20 μM). The enhanced anticancer activity of compounds **11**–**13** was further supported by their ability to induce apoptosis, as demonstrated by increased phosphatidylserine externalization, caspase activation, and DNA fragmentation. Additionally, an in vitro metabolic stability assay suggested potential oxidation pathways for these compounds. These findings highlight the potential of benzenesulfonamide–imidazole hybrids as promising candidates for the development of new and effective anticancer agents. However, further research is necessary to optimize their design, enhance their efficacy and ensure their safety.

## Data Availability

All data are available as Appendix A.

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
