# Peer review of "Novel 2-alkythio-4-chloro-N-[imino(heteroaryl)methyl]benzenesulfonamide Derivatives: Synthesis, Molecular Structure, Anticancer Activity and Metabolic Stability"

_ijms, 2023, doi:10.3390/ijms24119768_

Round 1

Reviewer 1 Report

1. In the introduction section, current data on cancer epidemiology should be provided insted of these in 2020.

2. It would be great if the Authors stated what type of hybrids were used, i.e., merged, fused etc. This information should also be introduced in the introduction. There are several papers with such info, please see:

https://pubmed.ncbi.nlm.nih.gov/27538458/; https://www.mdpi.com/1422-0067/23/7/3739; https://doi.org/10.1021/jm058225d; https://link.springer.com/chapter/10.1007/128_2010_76; 

In line with this some structures used in the study should be marked, and building pharmacophores should be distinguished.

3. Please provide information regarding compounds purity (any HPLC?)

4. In the methodology section there are some mistakes with numbering; 3.1. is for the synthesis while 3.2 should be for the Xray structure determination, no?

5. The subtitle "Metabolic stability" should be replaced with that which corresponds to the method, i.e., chemical or enzymatical stability.

6. Chemical stability refers to compounds exposed either to UV, acids, basic, temperature etc. The withdrawal of NADPH from the incubation mixture is not a method to perform chemical stability.

7. The Authors explained the reason to use breast, cervical and colon cancer cell lines, as these types are the most problematic these days. However, I'm wondering whether the chemical composition of the hybrid(s) corresponds to a particular type of cancer used, as some clinically available drugs are useful for specific types, and not for every cancer

8. The Authors should compare the results (here cytotoxicity, IC50) with compounds widely used for the therapy of a specific cancer type. Otherwise, it is hard to see the potential of the compounds.

9. It can be seen that the cytotoxic effect induced by the compounds selected is both concentration- and time-dependent. I was wondering whether the Authors have additional info regarding pH and temeprature of incubation impact? It could improve the paper greatly.

10. Whta was the reason to use for further studies only HeLa cells? In my opinion additional studies should also be performed for HaCaT and the compound 11-13.

11. The Authors stated that there were in vivo studies (please see line 247). Please correct this, as the "metabolic stability analysis" was in vitro.

12. What were the potential metabolits of the compound used to study the metabolism?

13. There is no discussion!

14. Conclusion should be short and specific

moderate correction is required

Author Response

Dear Sir,

The authors are very thankful for constructive comments and suggestions that have ultimately improved this manuscript. The manuscript have been modified according to the reviewer’s suggestions. All revisions in manuscript are marked in red. Detailed responses are given below.

In the response to Reviewer #1:

  1. In the introduction section, current data on cancer epidemiology should be provided insted of these in 2020.

Data on cancer epidemiology, indicated in manuscript, based on information provided by the World Health Organization (WHO). The latest available epidemiological data on the number of deaths worldwide due to cancer is from 2020. The Raport of International Agency for Research on Cancer; 2020 is officially cited on the WHO website.

There is known an article CA: A Cancer Journal for Clinicians, Cancer statistics, 2023, Rebecca L. Siegel MPH, Kimberly D. Miller MPH, Nikita Sandeep Wagle MBBS, MHA, PhD, Ahmedin Jemal DVM, PhD, https://doi.org/10.3322/caac.21763 that has reported the number of cancer deaths expected to occur in 2023. However the results were estimated by applying the previously described data-driven joinpoint algorithm to reported cancer deaths from 2006 through 2020 at the state and national levels as reported by the National Center for Health Statistics.

  1. It would be great if the Authors stated what type of hybrids were used, i.e., merged, fused etc. This information should also be introduced in the introduction. There are several papers with such info, please see:

https://pubmed.ncbi.nlm.nih.gov/27538458/; https://www.mdpi.com/1422-0067/23/7/3739; https://doi.org/10.1021/jm058225d; https://link.springer.com/chapter/10.1007/128_2010_76; 

In line with this some structures used in the study should be marked, and building pharmacophores should be distinguished.

We are grateful for your valuable comments. Information about types of hybrids was added to main text and suggested examples of literature were included as well.

For molecular hybrids described in our work the fused pharmacophores strategy was used and this information is introduced in introduction. We design the molecular hybrids by combining a 2-alkylthiobenzenesulfonamide fragment with heterocyclic systems such as imidazole, 1,2,4-triazole, and benzimidazole. This strategy involves linking two pharmacophors presented in many anticancer drugs by short linker or the pharmacophores are adjacent to each other.

According to the Reviewer’s suggestion we added in Figure 1 structure of target compounds (Formula A) with distinguished buildings pharmacophores.

  1. Please provide information regarding compounds purity (any HPLC?)

The purity of the compounds was assessed on the basis of elemental analysis. In addition, HPLC analysis was performed and results of purity assessment were added to the experimental section. Purity of compounds ranged from 91,23% to 99,76% .

HPLC-UV analysis was performed on Agilent 1260 liquid chromatograph equiped with VWD detector. Poroshell EC-C18 column (150 x 3 mm, 2.7 um) was used with the flow rate 0.2 ml/min. Injection volume was 5 μL. Gradient elution was applied as follows: linear increase of acetonitrile in water from 5% to 100% over 30 min. Detection was performed at 254 nm.

  1. In the methodology section there are some mistakes with numbering; 3.1. is for the synthesis while 3.2 should be for the Xray structure determination, no?

We are grateful for bringing attention to the mistake. Mistake has been corrected

  1. The subtitle "Metabolic stability" should be replaced with that which corresponds to the method, i.e., chemical or enzymatical stability.

We greatly appreciate the reviewer's  to align the subtitle with the specific method employed, thereby enhancing the clarity of our study. Taking this into account, we propose revising the subtitle "Metabolic stability" to "In vitro metabolic stability assay." This modification accurately reflects the methodology employed in our study, where we conducted experiments using an in vitro enzymatic system with microsomes and NADPH.

By adopting the revised subtitle, we aim to provide readers with a clearer understanding of the experimental approach utilized to assess the compounds' stability. This adjustment ensures that the terminology employed aligns precisely with the methods undertaken, thereby eliminating any potential confusion.

  1. Chemical stability refers to compounds exposed either to UV, acids, basic, temperature etc. The withdrawal of NADPH from the incubation mixture is not a method to perform chemical stability.

We appreciate the reviewer's observation regarding the incorrect use of the term "chemical stability" and agree with the suggestion for clarification. For compounds studied in another study, we did observe a decrease in the concentration of certain derivatives that was independent of the presence of NADPH. It is important to note that various physical and chemical phenomena can contribute to this observation, such as solubility problems and degradation, particularly hydrolysis.

Considering the reviewer's comment, we concur that a more appropriate term to describe this phenomenon would be a "negative control reflecting NADPH-independent processes, such as chemical degradation and precipitation." This revised terminology accurately captures the nature of the experiments conducted and provides readers with a clearer understanding of the specific processes under investigation.

  1. The Authors explained the reason to use breast, cervical and colon cancer cell lines, as these types are the most problematic these days. However, I'm wondering whether the chemical composition of the hybrid(s) corresponds to a particular type of cancer used, as some clinically available drugs are useful for specific types, and not for every cancer

In our research we took into account the fact that imidazole, triazole rings correspond to anticancer activity against colorectal cancer (dacarbazine) and breast cancer (anastrozole, letrozole, vorozol), sulfonamide pazopanib is used to cervical cancer treatment.

In addition, the search for new compounds with cytotoxic activity against cancer cells was guided by structural similarity to known sulfonamide and heterocyclic compounds with verified anticancer activity. Our goal was not to design a compound specific to a particular molecular target which, by binding to this target, induces apoptosis of cancer cells. The goal was to obtain a compound with pro-apoptotic activity and this goal was achieved. In further research, we will look for a molecular target for the obtained sulfonamides.

  1. The Authors should compare the results (here cytotoxicity, IC50) with compounds widely used for the therapy of a specific cancer type. Otherwise, it is hard to see the potential of the compounds.

We compared our results with cisplatin which is common drug used for cervical cancer treatment in clinical as a cell cycle non-specific drug. Information about this comparition was added to section 2.2.  

  1. It can be seen that the cytotoxic effect induced by the compounds selected is both concentration- and time-dependent. I was wondering whether the Authors have additional info regarding pH and temeprature of incubation impact? It could improve the paper greatly.

As observed by the Reviewer, the cytotoxic effects of the examined compounds were both time- and concentration-dependent. This was associated with the induction of cell death and the time-dependent activation of events involved in the execution of the apoptotic process. The experiments were carried out with the use of media optimized for the growth of the examined cell lines and in an incubator with controlled conditions of 5% C02 and a temperature of 370C, optimal for the growth of cells. Changing these parameters would hinder the growth of the cells and would enable examining the antiproliferative effects of the compounds. Thus, changes to the pH of the media or incubation temperature would not benefit this study.

  1. Whta was the reason to use for further studies only HeLa cells? In my opinion additional studies should also be performed for HaCaT and the compound 11-13.

The reason for selecting HeLa cells for further experiments was associated with the highest sensitivity of this cell line toward the compounds. HaCaT cells were used as a non-cancerous control for the examination of the toxicity of the compounds. Since the activity of compounds was higher toward the cancer cells with a selectivity index of 3, the determination of the mechanism of activity induced by the compounds was evaluated only toward the cancer cells.

  1. The Authors stated that there were in vivo studies (please see line 247). Please correct this, as the "metabolic stability analysis" was in vitro.

We are grateful to the Reviewer for pointing out this mistake. We corrected it.

  1. What were the potential metabolits of the compound used to study the metabolism?

Our intention was to study metabolic stability as one of the main indicators of a good drug candidate. There were no metabolites used in our study. Our in vitro experiment generated them, but we only followed the decrease of parent compound concentrations. However, based on in silico tools described in the manuscript we were able to describe possible metabolic pathways. The most probable metabolites could be products of oxidation such as aromatic, aliphatic oxidation or S-oxidation based on in silio prediction.

  1. There is no discussion!

We developed discussion and added proper phrases in red in section “Results and discussion”

  1. Conclusion should be short and specific

Conclusions have been shortened.

Reviewer 2 Report

The article "Novel 2-alkythio-4-chloro-N-[imino(heteroaryl)methyl]benzenesulfonamide derivatives: synthesis, molecular structure, anticancer activity and metabolic stability" is an interesting work presenting new benzenesulfonamide derivatives as anticancer agents against multiple malignant cell types. The manuscript is well written and organized, results are clearly presented and the experimental part is adeguately described.

Hence, the article is suitable for publication after the following minor revisions:

- Similarly to sulfonamides, add a figure representing the imidazoles (nilotinib) and triazoles (anastrozole, letrozole, etc.) described in the introduction

- In scheme 1, add reagents and conditions in the caption

- In paragraph 2.2 add the range of concentrations used for MTT

- At the end of line 43, no literature references are reported. Add examples of recent studies about the search of new chemotherapeutics: Archives of Pharmacal Research 45(11):1-16 DOI: 10.1007/s12272-022-01414-1; J Enzyme Inhib Med Chem. 2022 Dec;37(1):866-875. doi: 10.1080/14756366.2021.2014830; Molecules. 2022 Nov 2;27(21):7485. doi: 10.3390/molecules27217485; Int. J. Mol. Sci. 2022, 23, 10222. https://doi.org/10.3390/ijms231810222

- In the experimental part, several C13 NMR are missing. Please add them to even out the section.

Author Response

Dear Sir,

The authors are very thankful for constructive comments and suggestions that have ultimately improved this manuscript. The manuscript have been modified according to the reviewer’s suggestions. All revisions in manuscript are marked in red. Detailed responses are given below.

In the response to Reviewer 2

1. Similarly to sulfonamides, add a figure representing the imidazoles (nilotinib) and triazoles (anastrozole, letrozole, etc.) described in the introduction

We are grateful to the Reviewer for pointing out the missing structures in Figure. We added proper structures to Fig. 1

2. In scheme 1, add reagents and conditions in the caption

Reagents and conditions were added in the caption.

3. In paragraph 2.2 add the range of concentrations used for MTT

Compounds were examined in the concentration range 1, 10, 25, 50 and 100 μM. Missing information was added to section 2.2.

4. At the end of line 43, no literature references are reported. Add examples of recent studies about the search of new chemotherapeutics: Archives of Pharmacal Research 45(11):1-16 DOI: 10.1007/s12272-022-01414-1; J Enzyme Inhib Med Chem. 2022 Dec;37(1):866-875. doi: 10.1080/14756366.2021.2014830; Molecules. 2022 Nov 2;27(21):7485. doi: 10.3390/molecules27217485; Int. J. Mol. Sci. 2022, 23, 10222. https://doi.org/10.3390/ijms231810222

We are grateful for your valuable comments. Suggested examples of literature are included in the main text.

5. In the experimental part, several C13 NMR are missing. Please add them to even out the section.

We added all missing 13C NMR.

Round 2

Reviewer 1 Report

The Authors have now provided sufficient corrections and explanations to the text Therefore, in my opinion, it is ready for publishing.

minor changes are required